# Association of Oral Health with Multimorbidity among Older Adults: Findings from the Longitudinal Ageing Study in India, Wave-1, 2017–2019

**DOI:** 10.3390/ijerph182312853

**Published:** 2021-12-06

**Authors:** Srikanta Kanungo, Shishirendu Ghosal, Sushmita Kerketta, Abhinav Sinha, Stewart W Mercer, John Tayu Lee, Sanghamitra Pati

**Affiliations:** 1Division of Public Health, ICMR-Regional Medical Research Centre, Department of Health Research, Bhubaneswar 751023, India; srikantak109@gmail.com (S.K.); shishirendu123@gmail.com (S.G.); sushmita.kerketta07@gmail.com (S.K.); 2Health Technology Assessment in India (HTAiN), ICMR-Regional Medical Research Centre, Department of Health Research, Bhubaneswar 751023, India; dr.abhinav17@gmail.com; 3Usher Institute, College of Medicine and Veterinary Medicine, University of Edinburgh, Edinburgh EH16 4TJ, UK; Stewart.Mercer@ed.ac.uk; 4The Nossal Institute for Global Health, Melbourne School of Population and Global Health, The University of Melbourne, Melbourne, VIC 3010, Australia; johntayulee@unimelb.edu.au; 5Public Health Policy Evaluation Unit, Department of Primary Care and Public Health, School of Public Health, Imperial College London, London SW7 2AZ, UK

**Keywords:** ageing, India, multimorbidity, oral health, LASI Wave-1

## Abstract

India is witnessing an increase in the prevalence of multimorbidity. Oral health is related to overall health but is seldom included in the assessment of multimorbidity. Hence, this study aimed to estimate the prevalence of oral morbidity and explore its association with physical multimorbidity using data from Longitudinal Ageing Study in India (LASI). LASI is a nationwide survey amongst adults aged ≥ 45 years conducted in 2018. Descriptive analysis was performed on included participants (*n* = 59,764) to determine the prevalence of oral morbidity. Multivariable logistic regression assessed the association between oral morbidity and physical multimorbidity. Self-rated health was compared between multimorbid participants with and without oral morbidity. Oral morbidity was prevalent in 48.56% of participants and physical multimorbidity in 50.36%. Those with multimorbidity were at a higher risk of having any oral morbidity (AOR: 1.60 (1.48–1.73)) than those without multimorbidity. Participants who had only oral morbidity rated their health to be good more often than those who had physical multimorbidity and oral morbidity (40.84% vs. 32.98%). Oral morbidity is significantly associated with physical multimorbidity. Multimorbid participants perceived their health to be inferior to those with only oral morbidity. The findings suggest multidisciplinary health teams in primary care should include the management of oral morbidity and physical multimorbidity.

## 1. Introduction

India is currently witnessing a demographic transition with a rapid rise in the ageing population, due to socio-economic and healthcare advancements in the last two decades [1]. Projections suggest that the number of adults aged 60 years and above in India will rise to around 319 million by 2050 which will be approximately 20% of the national population [2]. This group has complex social, health, and economic needs which will have a major impact on the already overburdened healthcare delivery system.

Oral morbid conditions, such as dental caries and periodontitis, are one of the major public health challenges worldwide amongst disadvantaged people, especially in low- and middle-income countries (LMICs) [3]. The ‘Global Burden of Oral Conditions in 1990–2010’, derived from The Global Burden of Disease (GBD) 2010 study, suggests 3.9 billion people have oral morbidities globally [4]. The factors which directly contribute to poor oral health are lifestyle-related risk factors, along with limited availability and accessibility of services for oral health [5]. Additionally, chronic conditions may be related, as both share common risk factors [6]. Studies have described a bi-directional relationship between diabetes and periodontitis, i.e., people with diabetes often have severe gum disease and vice versa [7]. Periodontal diseases are also associated with increased risk of heart disease, stroke, and respiratory conditions such as pneumonia and chronic obstructive pulmonary disease (COPD) [8,9].

The term multimorbidity refers to the existence of two or more long-term conditions in a single individual [10]. These chronic conditions can either be non-communicable diseases (NCDs) of long duration such as hypertension, long-term mental health conditions such as a mood disorder, or infectious diseases of chronic nature. The prevalence of multimorbidity increases with age; most elderly people develop several NCDs and other health challenges such as degenerative changes, frailty, and mental and cognitive disorders [11]. In LMICs such as India, the demographic transition has increased the prevalence of multimorbidity as estimated by our previous study in primary care which identified multimorbidity to be common among older people, with the prevalence varying from 25% to 44.4% among adults aged 45 years and above [12].

Despite the increasing prevalence of multimorbidity, little is known about the association between oral morbidities and physical multimorbidity in India. Even though multimorbidity is increasing, current interventions and clinical practices commonly focus only on single physical conditions rather than multimorbidity including oral conditions [13]. However, the recent Ayushman Bharat scheme envisages providing basic oral healthcare through health and wellness centres being established throughout the country. With increasing life expectancy, policymakers need evidence-based information to understand the need to address both oral issues while focusing on other chronic conditions simultaneously. Hence, this study was carried out to address the present research gap with an aim to estimate the prevalence of oral morbidities and investigate correlates of oral health conditions and explore their association with physical multimorbidity among older adults aged 45 years and above in India using nationally representative data from Longitudinal Ageing Study in India (LASI), Wave-1.

## 2. Materials and Methods

### 2.1. Data Source

This study was based on the data from the Longitudinal Ageing Study in India (LASI), Wave-1, a nationwide survey to scientifically investigate health and its social determinants and related economics. It was conducted by Harvard TH Chan School of Public Health, the International Institute for Population Sciences (IIPS), and the University of Southern California. The data collection across the country spanned from April 2017 to December 2018 following a multistage stratified area probability cluster sampling to identify the unit of observation, i.e., LASI-eligible household (LEH) comprising one or more persons residing in a house with a cooking facility with at least one member aged ≥45 years. Data were collected from all 29 states and 6 Union territories of India following a three-stage sampling design for rural areas and a four-stage sampling for urban areas (Section A.1). Eligible candidates from selected households were given a survey instrument that had three segments—namely, Household Survey Schedule, Individual Survey Schedule, and Community Survey Schedule. Information for the Household Survey Schedule was collected from one or more adults who were aware of the socio-demographic status, financial details, water supply, sanitation, etc. of the household. Through the Individual Survey Schedule, the details of the consenting older adult (aged ≥ 45 years) and their spouses irrespective of age (if applicable) were recorded. The detailed methods of LASI Wave-1 have been reported on the website of the International Institute for Population Sciences (IIPS), Mumbai [14].

### 2.2. Study Participants and Sample Size

During the survey, 44,462 age-eligible households were identified, and among them, 72,250 individuals completed the interview from 42,949 households. We excluded participants with missing/wrongly entered/invalid data based on the outcome or independent variables that were used for analysis. Following this, 59,764 participants aged ≥ 45 years formed our study population. In India, the onset of NCDs is a decade earlier than in the high-income countries due to which this age group was selected for this study [15].

### 2.3. Ethical Considerations

LASI received ethical clearance from the Indian Council of Medical Research (ICMR) and IIPS, Mumbai. This study was based on anonymous secondary data obtained from LASI Wave-1, and hence, there was no risk to participation. The data were requested from IIPS, Mumbai, through proper channels, and appropriate permission was taken. The same has been properly acknowledged and referenced wherever required.

#### 2.3.1. Variables

##### Oral Morbidity

The Individual Survey Schedule consisted of self-reported health conditions, along with other socio-demographic details. Participants were asked if they were diagnosed with any one or multiple of the seven specific oral conditions such as, ‘painful teeth’, ‘ulcers lasting more than two weeks’, ‘bleeding gums’, ‘swelling gums’, ‘loose teeth’, ‘dental cavity/dental caries’, ‘soreness or cracks in the corner of the mouth’ and/or any other conditions in the last one year. Other conditions were reported by 51 participants which were ‘not good teeth’, ‘germs problem’, ‘root canal’, ‘heaviness in tongue’, ‘mouth problem’, etc. which was irrelevant, hence were not included in the analysis. We segregated the responses and stratified them into four categories—‘none’, ‘one oral condition’, ‘two oral conditions’ and ‘>two oral conditions’—to find out the prevalence of oral conditions across different socio-demographic attributes. Further, we classified oral conditions based on soft tissue and hard tissue conditions (Section A.4). 

### 2.4. Exposure Characteristics

#### Socio-Demographic Characteristics

This included information of the respondents such as age, gender, residence (rural/urban), caste (four categories), educational qualification (five categories), employment status (currently working and not working), wealth index (five quantiles), and if the respondent is currently living with his/her partner. 

For analysis, we grouped age into three categories, i.e., ‘45–59 years’, ‘60–74 years’, and ‘75 years and above’. We stratified educational qualification into four categories as follows: ‘no formal education’ for those who never attended school; ‘up to primary school’, combining two groups of ‘less than primary’ and ‘primary completed’; the next group was ‘middle school to higher secondary’ that consisted of three groups of ‘middle school completed’, ‘secondary school/matriculation completed’ and ‘higher secondary/intermediate/senior’; and lastly, all the remaining groups, ‘diploma and certificate holders’ ‘graduate degree’, ‘post-graduate degree’, and ‘professional course/degree’ were added together for our final category, ‘diploma, graduate, and above’. Further, participants were grouped based on their marital status, who had a life partner (‘currently married’ and ‘live in a relationship’) and those who did not have (‘divorced’ + ‘separated’ + ‘deserted’ + ‘widowed’ + ‘never married’) presently. States and UTs of the country were divided into six regions east, west, north, south, and northeast. First, three strata of ‘caste’ were adopted as it was from the data. A new category, ‘others’ was created merging ‘none of them’ and ‘no caste/tribe’. Individuals, who were ‘not working currently’ and ‘did not work for a minimum of 3 months in a lifetime’ were treated as ‘currently not working’, whereas ‘currently working’ members were labelled as in the original data. Participants were asked to rate their health in general, and the answer of each individual was on a five-point Likert scale mentioned as excellent, very good, good, fair, and poor. 

### 2.5. Multimorbidity

During data collection, details of 12 types of self-reported chronic physical diseases and conditions (hypertension, diabetes, cancer, chronic lung disease, chronic heart disease, stroke, arthritis and osteoporosis or other bone/joint diseases, neurological or psychiatric problems, hypercholesterolemia, thyroid disease, gastrointestinal problems, chronic renal disease) and four other morbid conditions (skin diseases, vision and hearing defect, and obesity) were collected. Obesity was assessed on the basis of body mass index (BMI) calculated from height and weight using reference cutoff for the Asian population [16]. In alignment with our objective, we categorised multimorbidity into two groups as physical multimorbidity absent (those who did not have any of the conditions or had only one condition) and physical multimorbidity present (those who had two or more conditions). Additionally, we have also classified these chronic conditions on the basis of body systems following chapters from the International Classification of Diseases (ICD)-10 (Section A.4). 

### 2.6. Data Management and Analysis

We calculated the mean and standard deviation for the continuous data (age). We also analysed profiles of various diseases among both genders and expressed them in numbers and proportions. Distribution of the number of oral morbidities in relation to other categorical variables (such as age group, gender, area of residence, educational qualification, life partner, caste, employment status, national region, and wealth index) was presented as frequencies (n, n%) and *p*-value. *p*-value less than 0.05 was considered statistically significant. Multivariate logistic regression was performed, expressed as adjusted odds ratio (AOR) with 95% confidence interval (CI). We separately analysed the proportion of multimorbid participants with and without oral conditions, rating their health status expressed with a significance value obtained from the Mantel Haenszel chi-square test.

## 3. Results

The number of included participants was 59,764 with an age range from 45 years to 116 years and a mean age of 60.22 (± 10.64) years (Table 1). Almost two-thirds of the participants were from rural areas (69.87%), with just about one-third (30.13%) from urban areas. Almost half (49.79%) of the participants were 45–59 years of age. Participants from different socio-economic groups were almost equally distributed. Gender distribution was also almost equal, with 45.86% males and 54.14% females. Almost half (48.56%) of the participants had at least one oral morbidity, whereas a higher number of participants (50.36%) had physical multimorbidity. The prevalence and association of oral morbidity with various socio-demographic correlates are described in the Section A.2. The detailed distribution of various physical chronic conditions across gender is described in the Section A.3.

Multivariate regression analysis showed a higher probability of having any oral morbidity among those 60–74 years old than 75 years and above, with AOR 1.27 (1.18–1.37) and 1.02 (0.90–1.16), respectively, as compared with the participants aged 45–59 years (Table 2). Similar chances of having oral morbidity were observed among the rural population (AOR 1.06 (0.97–1.15)), as compared with their urban counterparts. There was a small difference in having oral morbidity among females (AOR 1.13 (1.05–1.22)), as compared with males. The highest likelihood of having oral morbidity (AOR 1.22 (1.09–1.38)) was reported among the most affluent class, as compared with the participants of other wealth quantiles. 

A higher likelihood of having oral morbidity was found to be associated with lesser educational qualification. Respondents who never went to school had higher odds (AOR 1.84 (1.44–2.36)) of having oral morbidity, as compared with those having diplomas, graduate degrees, and above. We found that the prevalence of oral morbidity increased with the presence of multimorbidity, and there was a higher likelihood (AOR 1.60 (1.48–1.73)) of having oral morbidity in patients who had physical multimorbidity, as compared with those who did not have multimorbidity. Similarly, increased numbers of physical chronic conditions within individuals were associated with a higher likelihood of having oral morbidities (Figure 1). A high proportion (16.6%) of the participants who had two physical chronic conditions reported to have two oral morbidities which increased to 17.8% among participants with three or more chronic conditions. Multimorbid individuals had a greater burden of oral morbidity which also tended to rise with an increase in the number of physical chronic conditions. 

A higher number of participants who had only oral morbidity rated their overall health to be excellent (3.87%), very good (18.38%), and good (40.84%) than those who had physical multimorbidity along with oral morbidity, who rated their health to be excellent (2.32%), very good (11.62%), and good (32.98%) (Table 3). Further, we described the frequency of oral conditions grouped as soft tissue and hard tissue across physical conditions classified on the basis of various body systems (Section A.4). We found soft tissue conditions were common among patients with circulatory conditions (40.14%), whereas hard tissue conditions were common among individuals with conditions of endocrine, nutritional, and metabolic conditions (40.45%).

## 4. Discussion

Using the nationally representative data of India, we found a high prevalence of physical multimorbidity and oral morbidity among older adults. A key finding of this study was the increased chance of having oral morbidity among multimorbid individuals, who also rated their health to be compromised more, as compared with their counterparts who did not have oral morbidity. Our study estimated a 48% prevalence of oral morbidity among adults aged 45 years and above, but various studies have reported different prevalence, depending on the age group and number of oral conditions observed. A similar study conducted in New Delhi reported a 91.9% prevalence of dental caries among adults aged ≥60 years [17] which is higher than the findings of our study; conversely, in contrast to our findings, a study conducted among rural Indian adults aged 35–54 years reported a 13% prevalence of one or more oral health impacts [18] which is considerably lower than the present study. However, there is a paucity of comprehensive data reporting the exact extent of the burden of oral morbidities in India which could mainly be attributed to considering oral health of almost trivial value. 

Ageing often increases risk factors such as cognitive and functional impairment, reduced mobility, and frailty, which hampers the quality of life leading to dependence [19]. These conditions are directly associated with compromised oral health [20]. There has been a substantial increase in multimorbidity among LMICs such as India, mainly attributed to the rise in non-communicable diseases alongside prevailing chronic infectious diseases. Our previous systematic review assessed the prevalence of multimorbidity in South Asia (4.5% to 20.8%) among adults aged 18 years and above which increased among older adults aged 60 years and above (24.1% to 83%) [21]. This indicated multimorbidity to be more akin among the ageing population, hence escalating the risk for compromised oral health. Our study reported 50.36% of the older adults to have multimorbidity which is consistent with the findings of a multinational study (including India) reporting 56.6% multimorbidity among adults aged ≥50 years [22]. These long-term chronic conditions of 10 present complex-care needs put a financial burden on the individual; as a result, oral health needs are often overshadowed both for the patients and their caregiving family members [23], as shown by the findings of our study which revealed oral morbidities to be commonly associated with a certain group of physical conditions such as endocrine disorders, visual impairment, and circulatory system diseases. A probable reason for the high frequency of oral conditions among individuals with visual impairment can be their inability to maintain oral hygiene. Further, circulatory system diseases are known to be associated with bleeding and swelling gums (soft tissue) and respiratory conditions [8,9].

A similar study to investigate differences in the distribution of oral health indicators among elderly people with and without multimorbidity observed a significant difference in the prevalence of edentulism among multimorbid participants than those without multimorbidity (53.3% vs. 32.7%. *p* = 0.015) [24] which is consistent with the findings of this study. Inextricably, oral health is related to overall health and quality of life but is seldom included in the assessment of multimorbidity, leading to the scarcity in the available literature. Here, it is worth noting that the available studies were confined to multimorbidity excluding oral conditions, posing a challenge to compare our findings with other similar studies. 

Self-rated health (SRH) is regarded as a proxy indicator for quality of life. We found a significant proportion of multimorbid individuals who had oral morbidity rated their health to be poorer than those who did not have any oral morbidity. This could be due to the complex correlation between physical multimorbidity and shared risk factors of ageing which compels individuals to polypharmacy. Polypharmacy further cascades to frailty through weight loss, and exhaustion has poor outcomes such as disability [25]. Furthermore, all these factors club to have an ill effect on general and oral health, leading to decreased self-esteem and affecting mental health and quality of life. A study conducted to find the association between mental and oral health in Spain reported a positive association between the presence of oral conditions and having at least one psychiatric condition [26].

### 4.1. Implications for Policy and Practice

Most of the oral conditions are preventable and share risk factors with other physical chronic conditions. Hence, an aim to reduce risk factors such as frequent snacking, sugar consumption, tobacco, and alcohol consumption should be targeted. Awareness regarding oral hygiene should be created from childhood. People should be motivated to equally prioritise oral health similar to other physical chronic conditions, so as to prevent extensive damage and reduce economic burden. Additionally, coordinated quality care as provided by health and wellness centres, under one umbrella, should be aimed. Multidisciplinary teams at primary care should holistically approach patient-centred treatment. Further, longitudinal studies on multimorbidity including oral morbidities are needed to generate evidence on clustering of oral conditions among multimorbid individuals.

### 4.2. Strengths and Limitations

To the best of our knowledge, this is the first study in India to investigate the association of oral morbidity with physical multimorbidity. The main strong point of our study lies in the use of nationally representative data captured using rigorous scientific methodology. LASI takes into account self-reported health conditions which were used to determine physical multimorbidity, a limitation for this study. Various correlates such as smoking, alcohol, and sugar intake which affect both oral and general health could not be analysed due to insufficient data. 

## 5. Conclusions

This study suggested a high burden of oral morbidity which is significantly associated with multimorbidity. Furthermore, multimorbid participants with oral morbidity perceived their health to be more compromised than those having only oral morbidity. The findings of this study suggest the inclusion of oral healthcare services in primary care settings while managing physical multimorbidity. Additionally, future studies to establish linkages between physical multimorbidity and oral morbidity are warranted.

## Figures and Tables

**Figure 1 ijerph-18-12853-f001:**
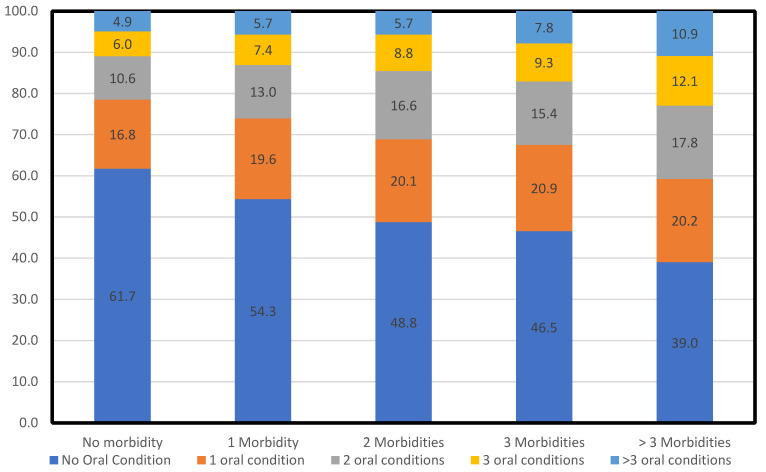
Relationship between count of multimorbidity and oral morbidity.

**Table 1 ijerph-18-12853-t001:** Socio-demographic characteristics of study population.

Attributes	*n* (%)
**Age (*n* = 59,764)**	
**Mean (± SD):** 60.22 (±10.64) years, **Range:** 45 to 116 years.	
45–59	29,756 (49.79)
60–74	23,539 (39.39)
≥75	6469 (10.82)
**Gender (*n* = 59,764)**	
Male	27,405 (45.86)
Female	32,359 (54.14)
**Residence (*n* = 59,764)**	
Rural	41,759 (69.87)
Urban	18,005 (30.13)
**Regions of India (*n* = 59,764)**	
North	4649 (7.78)
Central	15,461 (25.87)
East	14,180 (23.73)
Northeast	2054 (3.44)
West	9452 (15.82)
South	13,968 (23.37)
**Caste (*n* = 59,275)**	
SC	11,547 (19.48)
ST	5113 (8.63)
OBC	27,162 (45.83)
Others	15,452 (26.07)
**Educational Qualification (*n* = 59,763)**	
No formal Education	30,375 (50.82)
Up to Primary school	13,925 (23.30)
Middle school to Higher Secondary	12,342 (20.65)
Diploma, Graduate, and Above	3122 (5.22)
**Employment Status (*n* = 59,760)**	
Currently working	27,926 (46.73)
Currently not working	31,834 (53.27)
**Life Partner (*n* = 59,762)**	
Have a partner	44,326 (74.17)
Does not have a partner	15,436 (25.83)
**Wealth Index (*n* = 59,764)**	
The most deprived quantile	12,615 (21.11)
2nd quantile	12,715 (21.27)
3rd quantile	12,210 (20.43)
4th quantile	11,725 (19.62)
The most affluent quantile	10,499 (17.57)

**Table 2 ijerph-18-12853-t002:** Multivariate analysis between oral morbidity and its correlates.

Correlates		Oral Morbidity	
Morbidity	Categories	AOR (95% Confidence Interval)	*p*-Value
Age (*n* = 59,764)	45–59	Reference	
	60–74	1.27 (1.18–1.37)	<0.001
	≥75	1.02 (0.90–1.16)	0.701
Gender (*n* = 59,764)	Male	Reference	
	Female	1.13 (1.05–1.22)	<0.001
Residence (*n* = 59,764)	Rural	1.06 (0.97–1.15)	0.216
	Urban	Reference	
Regions of India (*n* = 59,764)	North	1.38 (1.26–1.52)	<0.001
	Central	1.19 (1.09–1.30)	<0.001
	East	1.52 (1.39–1.67)	<0.001
	Northeast	1.21 (1.09–1.34)	<0.001
	South	2.00 (1.77–2.26)	<0.001
	West	Reference	
Caste (*n* = 59,275)	SC	Reference	
	ST	1.06 (0.95–1.18)	0.298
	OBC	1.05 (0.97–1.14)	0.192
	Others	1.00 (0.92–1.09)	0.980
Educational Qualification (*n* = 59,763)	No formal Education	1.84 (1.44–2.36)	<0.001
	Up to Primary school	1.71 (1.33–2.20)	<0.001
	Middle school to Higher Secondary	1.49 (1.14–1.94)	0.004
	Diploma, Graduate, and Above	Reference	
Employment Status (*n* = 59,760)	Currently working	0.93 (0.86–1.00)	0.052
	Currently not working	Reference	
Life Partner (*n* = 59,762)	Have a Partner	1.06 (0.97–1.16)	0.185
	Do not have a Partner	Reference	
Wealth Index (*n* = 59,764)	The most deprived quantile	Reference	
	2nd quantile	1.10 (1.01–1.20)	0.038
	3rd quantile	1.13 (1.02–1.25)	0.017
	4th quantile	1.12 (1.01–1.24)	0.036
	The most affluent quantile	1.22 (1.09–1.38)	0.001
Multimorbidity (*n* = 59,764)	Absent	Reference	
	Present	1.60 (1.48–1.73)	<0.001

**Table 3 ijerph-18-12853-t003:** Self-rated health among multimorbid participants with and without oral morbidity.

Self-Rated Health	Oral Morbidity (*n* = 59,745)				Mantel Haenszel Chi-Square Test
	Present		Absent		
	Multimorbidity Present *n*, % (CI)	Multimorbidity Absent *n*, % (CI)	Multimorbidity Present *n*, % (CI)	Multimorbidity Absent *n*, % (CI)	*p*-Value
Excellent	382, 2.32 (2.10–2.56)	486, 3.87 (3.53–4.22)	549, 4.03 (3.70–4.37)	1058, 6.20 (5.84–6.57)	<0.001
Very Good	1912, 11.62 (11.14–12.12)	2312, 18.38 (17.71–19.07)	2269, 16.64 (16.01–17.27)	3974, 23.27 (22.64–23.92)	
Good	5425, 32.98 (32.26–33.71)	5138, 40.84 (39.99–41.71)	4592, 33.66 (32.87–34.46)	7295, 42.72 (41.98–43.47)	
Fair	5960, 36.23 (35.50–36.98)	3636, 28.90 (28.12–29.71)	4354, 31.92 (31.13–32.71)	3794, 22.22 (21.60–22.85)	
Poor	2770, 16.84 (16.27–17.42)	1007, 8.01 (7.54–8.49)	1878, 13.76 (13.19–14.36)	955, 5.59 (5.25–5.95)	

## Data Availability

The dataset analysed during the current study is available in the LASI data repository at ICT, IIPS (https://iipsindia.ac.in/content/lasi-wave-i (accessed on 22 January 2021)).

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
