# Peer review of "Association of Oral Health with Multimorbidity among Older Adults: Findings from the Longitudinal Ageing Study in India, Wave-1, 2017–2019"

_ijerph, 2021, doi:10.3390/ijerph182312853_

Round 1

Reviewer 1 Report

Page 3, Line 96-98. This paragraph is confusuing. The authors explain they included participants of 60 years old and above, and 75 years old and above, but they did not explain the thershold of age. For example "participants aged 60 to 74<, participants of 75 to........, Later they explain all patients of >45 years old were included, but they did not explain the limit of age 45 years to <74. 

Sample size. I recommend to the authors do not repeat the 45 years old group or above. Please make a better redaction of the paragraph.

Page 3; line 116; and/or any other...., could it be possible to explain what other conditions were present? 

Page 3; line 118. Authors only made the groups of the diseases according to the conditions, one oral condition, two oral condition (one disease) ..., I think that make groups with one or two condition might not be suitable for this study, could it be possible to establish the groups of the diseases according to their origin such as metabolic disease, infectious/inflammatory diseases, hereditary diseases, neoplastic diseases, and developmental diseases? I consider that this organization will be more appropriate.  

 Results  

The results were interesting, the authors established the relationship of number of morbidities with variables. However, classifying some important disease as a multimorbidity, may be not suitable for the study. I recommend to classify the multimorbidity as a specific disease. These classifications may give more support to the results in order to make interesting the tables. Likert scale does not seem to be adequate to rate the real health status.  

 I recommend to reclassify the multimorbidity into types of diseases (Metabolic disease, infectious/inflammatory diseases, hereditary diseases, neoplastic diseases, and developmental diseases) or make a table in which the frequency (%) of multi morbidities related with oral condition are described. 

 Discussion  

Page 8, Line 235 to 238. "This indicates...." This paragraph is interesting. I suggest the authors to describe what types of multimorbidity were more prevalent in ageing population and to compare with the results described in the same paragraph. 

The discussion was interesting and well done, the authors adequately compare  their results with other studies and perform an explanation of the obtained results. However, I consider the discussion may improve if a specific classification of the multimorbidity and oral morbidity is included.  

Overall comments 

This study is interesting and well done, the authors make a great effort to do this manuscript. However, classifying as morbidity and multimorbidity might not be suitable for the study, which is reflected in the discussion. Despite this, it is a well done manuscript. 

Author Response

Page 3, Line 96-98. This paragraph is confusing. The authors explain they included participants of 60 years old and above, and 75 years old and above, but they did not explain the threshold of age. For example "participants aged 60 to 74<, participants of 75 to........, Later they explain all patients of >45 years old were included, but they did not explain the limit of age 45 years to <74. 

Author’s Response: We have now merged study participants and sample size together with more easy to comprehend text. 

Sample size. I recommend to the authors do not repeat the 45 years old group or above. Please make a better redaction of the paragraph.

Author’s Response: We have changed as suggested.

Page 3; line 116; and/or any other...., could it be possible to explain what other conditions were present? 

Author’s Response: In total there are 51 participants who reported oral conditions as “others”. We have not included them in analysis as these conditions were as follows: “not good teeth”, “germs problem”, “root canal”, “heaviness in tongue” and “mouth problem” etc. which was irrelevant. Further, in Page 3; line 116 we have only mentioned the question based on which this variable was created.

Authors only made the groups of the diseases according to the conditions, one oral condition, two oral condition (one disease) ..., I think that make groups with one or two condition might not be suitable for this study, could it be possible to establish the groups of the diseases according to their origin such as metabolic disease, infectious/inflammatory diseases, hereditary diseases, neoplastic diseases, and developmental diseases? I consider that this organization will be more appropriate.  

Author’s Response: Thank you so much for your suggestion. Here, our objective is to look at the prevalence of oral conditions and its association with multimorbidity. We considered only conditions (one, two etc.) and not the groups to give a broad picture of how multimorbidity and oral morbidities are associated, despite which patients get fragmented care. Additionally, multimorbidity defined as two or more chronic conditions seldom does not takes into account oral morbidities.  

However, we agree with the suggestion of reviewer following which we have divided oral conditions as soft tissue and hard tissue as it was the most feasible classification. We could not classify the given oral conditions as suggested (metabolic disease, infectious/inflammatory diseases, hereditary diseases, neoplastic diseases, and developmental diseases).

Results  

The results were interesting, the authors established the relationship of number of morbidities with variables. However, classifying some important disease as a multimorbidity, may be not suitable for the study. I recommend to classify the multimorbidity as a specific disease. These classifications may give more support to the results in order to make interesting the tables.

Likert scale does not seem to be adequate to rate the real health status.  

Author’s Response: We agree with the reviewer, but this being a secondary data analysis, its limitation is the availability of variables which compelled us to take only self-rated health as a proxy indicator of health status. Additionally, self-rated health is a widely used proxy indicator (mentioned in the manuscript) for health related quality of life. 

I recommend to reclassify the multimorbidity into types of diseases (Metabolic disease, infectious/inflammatory diseases, hereditary diseases, neoplastic diseases, and developmental diseases) or make a table in which the frequency (%) of multi morbidities related with oral condition are described. 

Author’s Response: Thank you so much for your suggestion. As suggested we have classified these chronic conditions in groups based on body systems following chapters of ICD-10. Further, we have added a supplementary table showing the distribution of oral conditions based on this classification of chronic conditions.

 Discussion  

Page 8, Line 235 to 238. "This indicates...." This paragraph is interesting. I suggest the authors to describe what types of multimorbidity were more prevalent in ageing population and to compare with the results described in the same paragraph. 

Author’s Response: Paragraph representing the same added.

The discussion was interesting and well done, the authors adequately compare their results with other studies and perform an explanation of the obtained results. However, I consider the discussion may improve if a specific classification of the multimorbidity and oral morbidity is included.  

Author’s Response: Paragraph representing the same added.

Overall comments 

This study is interesting and well done, the authors make a great effort to do this manuscript. However, classifying as morbidity and multimorbidity might not be suitable for the study, which is reflected in the discussion. Despite this, it is a well done manuscript. 

Author’s Response: Thank you so much for your time. We appreciate your efforts in giving valuable feedback.

Reviewer 2 Report

1. the article needs to be proofread by a native speaker. 
2. Please correct punctuation errors, double spaces, etc. 

Introduction: 
Introduces the reader to the topic of the paper very well. It provides comprehensive information on both epidemiological and theoretical backgrounds.

Materials and Methods:
Did the bioethics committee approve the study?
Statistical Analysis: - Please describe what statistical tests were used in the study.

Results: 
The results are presented in the form of text, figures, and tables. Present the results, their statistical analysis, and discussion in detail. 

Discussion and Conclusions:
The discussion accurately describes the results, presents the authors' analytical approach to the topic, and concludes. 

The article presents an interesting topic very well. In the reviewer's opinion, the paper with minor corrections can be accepted for publication in IJERPH. 

Author Response

  1. The article needs to be proofread by a native speaker. 

Author’s Response: The article has been proofread by native speakers and all grammatical mistakes have been edited.

  1. Please correct punctuation errors, double spaces, etc.

Author’s Response: We have tried to correct all such errors in the manuscript.

Introduction: 
Introduces the reader to the topic of the paper very well. It provides comprehensive information on both epidemiological and theoretical backgrounds.

Author’s Response: Thank you so much for your positive feedback.

Materials and Methods:
Did the bioethics committee approve the study?

Author’s Response: Yes, we have now added about LASI’s overall bioethics approval in the ethical considerations section. However, for this study we did not take bioethical approval as it is a secondary data analysis.

Statistical Analysis: - Please describe what statistical tests were used in the study.

Author’s Response: Thank you for pointing out this error. We have added additional tests (Mantel Haenszel Chi square test) in the statistical analysis section.

Results: 
The results are presented in the form of text, figures, and tables. Present the results, their statistical analysis, and discussion in detail. 

Author’s Response: Thank you for your suggestion. We have tried to keep the manuscript concise and focused, however, we have added few more details as suggested.

Discussion and Conclusions:
The discussion accurately describes the results, presents the authors' analytical approach to the topic, and concludes. 

Author’s Response: Thank you so much for your positive feedback.

The article presents an interesting topic very well. In the reviewer's opinion, the paper with minor corrections can be accepted for publication in IJERPH. 

Author’s Response: Thank you so much for your time. We appreciate your efforts in giving valuable feedback.

Reviewer 3 Report

Thanks for submitting your manuscript entitled, Association of oral health with multimorbidity among older adults: Findings from Longitudinal Ageing Study in India, 3 Wave-1, 2017-2019. Please see my comments below.

Abstract

Introduction

  • The aims of this study were to examine the prevalence of oral morbidities and investigate correlates of oral health conditions and explore their association with physical multimorbidity among older adults aged 45 years and above in India using nationally representative data from Longitudinal Ageing Study in India (LASI) Wave-1. Unfortunately, It is unclear to show the research gap. Oral health has been evidenced to be closely associated with morbidity, particularly NCDs. The authors need to explain how the study to be conceptualized.
  • The in-text citation should follow the Journal’s instruction.
  • The sampling recruitment is unclear.
  • How did
  • The authors did not explain why the population aged 45 or older were selected. The study focused on oral health and comorbidities, but the sampling was not considered with these two major focuses.

Ascertainment of outcomes and exposures

  • Any checklist to help evaluate oral morbidities?

What is “Exposure characteristics”? It’s better to follow the Journal’s format.

Multimorbidity

The authors only defined this term according to the numbers of diseases. What about the types of diseases? Any diseases were excluded? Who collected the data? Multiple people? If so, how to make sure the consistency?

Results:

Page 6, what is AOR? Must have full term at the first time

The discussion was well written

References

Should follow the journal’s format

Author Response

Thanks for submitting your manuscript entitled, Association of oral health with multimorbidity among older adults: Findings from Longitudinal Ageing Study in India, 3 Wave-1, 2017-2019. Please see my comments below.

 Abstract

Introduction

  • The aims of this study were to examine the prevalence of oral morbidities and investigate correlates of oral health conditions and explore their association with physical multimorbidity among older adults aged 45 years and above in India using nationally representative data from Longitudinal Ageing Study in India (LASI) Wave-1. Unfortunately, It is unclear to show the research gap. Oral health has been evidenced to be closely associated with morbidity, particularly NCDs. The authors need to explain how the study to be conceptualized.

Author’s Response: Thank you for your constructive feedback on this manuscript. Our study aims to explore the association between oral morbidity and physical multimorbidity to state that multimorbid individuals who suffer from two or more chronic conditions not only have physical conditions but also oral conditions. The implications of this paper would more focus on strengthening primary care and integrating oral services along with other medical services at all levels. Hence, our intent for the need for this study seems clear from the introduction.

  • The in-text citation should follow the Journal’s instruction.

Author’s Response: Thank you for pointing this out. We have now changed in-text citation as per journal’s requirement.

  • The sampling recruitment is unclear.

Author’s Response: Sampling recruitment has now been changed and presented in a much clear manner which will be easy to comprehend. Additionally, supplementary file also contains a figure depicting it.

  • How did
  • The authors did not explain why the population aged 45 or older were selected. The study focused on oral health and comorbidities, but the sampling was not considered with these two major focuses.

Author’s Response: LASI is a nationally representative survey among older adults aged 45 years and above and their spouses (irrespective of age). In India, onset of NCDs is 10 years earlier than the high income countries due to which 45 years age has been selected to evaluate the health of ageing population. Also, in introduction we have mentioned regarding high prevalence of multimorbidity among this age group estimated through our previous study (reference 12). Following the same, we included this age group for our study.

Ascertainment of outcomes and exposures

  • Any checklist to help evaluate oral morbidities?

Author’s Response: All oral conditions are self-reported on the basis of prior diagnosis. We have included all oral morbidities included in the survey. For the ease of understanding, we have now clubbed oral morbidities as soft tissue conditions such as bleeding gums, swelling gums etc. and hard tissue conditions such as dental caries based on which a new supplementary table showing the distribution of these conditions across physical conditions groups has been added.

What is “Exposure characteristics”? It’s better to follow the Journal’s format.

Author’s Response: We have now changed the term to variables following the journal’s format.

Multimorbidity

The authors only defined this term according to the numbers of diseases. What about the types of diseases? Any diseases were excluded? Who collected the data? Multiple people? If so, how to make sure the consistency?

 Author’s Response: Multimorbidity is defined on the basis of count of diseases whereas complex multimorbidity takes into account the types of diseases clubbed together. Here, our objective was to look at the association of oral morbidity with multimorbidity. However, we appreciate reviewer’s suggestion following which we have classified these chronic conditions in groups based on body systems following chapters of ICD-10. Further, we have added a supplementary table showing the distribution of oral conditions based on this classification of chronic conditions.

Here, we would clarify that to the best of our knowledge, we included all chronic conditions whose data was available in LASI.

Data was collected by multiple field staff who were trained thoroughly to maintain the consistency of data. However, we agree with the reviewer that this is a limitation of any nationally representative survey data which has been mentioned in the limitations. 

Results:

Page 6, what is AOR? Must have full term at the first time

Author’s Response: AOR was used in page 4, line 165 for the first time where full term is mentioned.

The discussion was well written.

Author’s Response: Thank you so much for positive feedback.

References

Should follow the journal’s format

Author’s Response: Thank you for pointing this out. We have now changed references as per journal’s requirement.

Thank you so much for your time. We appreciate your efforts in giving valuable feedback.

Round 2

Reviewer 1 Report

The manuscript has improved considerably in comparison to the prior version. Now is more understandable and better explained.

Please review line 227 "(40.14)" is it necessary use the percent sign

Author Response

The manuscript has improved considerably in comparison to the prior version. Now is more understandable and better explained.

Author's Response: Thank you so much for your positive feedback. 

Please review line 227 "(40.14)" is it necessary use the percent sign

Author's Response: Changed as suggested. Thank you